# The Effects of Deworming and Multiple Micronutrients on Anaemia in Preschool Children in Bangladesh: Analysis of Five Cross-Sectional Surveys

**DOI:** 10.3390/nu14010150

**Published:** 2021-12-29

**Authors:** Haribondhu Sarma, Kinley Wangdi, Md Tariqujjaman, Ratish Das, Mahfuzur Rahman, Matthew Kelly, Tahmeed Ahmed, Darren J. Gray

**Affiliations:** 1The National Centre for Epidemiology and Population Health, Australian National University, Canberra, ACT 2601, Australia; kinley.wangdi@anu.edu.au (K.W.); matthew.kelly@anu.edu.au (M.K.); darren.gray@anu.edu.au (D.J.G.); 2Nutrition and Clinical Services Division, International Centre for Diarrhoeal Disease Research, Dhaka 1212, Bangladesh; md.tariqujjaman@icddrb.org (M.T.); mahfuzur.rahman@icddrb.org (M.R.); tahmeed@icddrb.org (T.A.); 3Gungahlin Medical Practice, Hibberson St., Gungahlin, ACT 2912, Australia; srbhaban2005@gmail.com

**Keywords:** Bangladesh, anaemia, children, deworming, micronutrient powders

## Abstract

Anaemia is a major public health problem among children < 5 years of age in Bangladesh due to recurrent intestinal parasite infections. The aim of this study was to understand the association between combining deworming and MNP home fortification (MNP + Deworming) and the prevalence of anaemia among children < 5 years of age in Bangladesh. We used pooled data from five cross-sectional surveys and performed multivariable logistic regression and calculated crude and adjusted odds ratios (AORs) to quantify the association of anaemia with the exposure variables. A total of 9948 households were considered for this paper. In the unadjusted logistic regression, no significant association was detected between the effective MNP coverage and anaemia prevalence, but the associations were significant (*p* < 0.001) between the deworming and anaemia prevalence and between the MNP + Deworming condition and anaemia prevalence. In the adjusted model, children who were exposed to both deworming and effective MNP coverage were 30% (AOR 0.70; 95% CI 0.52, 0.94; *p* = 0.018) less likely to be anaemic compared with children who were unexposed to combined MNP + Deworming. The combined effects of deworming and MNP supplementation on the reduction in anaemia prevalence highlighted the importance of using integrated and multidisciplinary intervention strategies.

## 1. Background

Globally, approximately 40% of children aged 6–59 months suffer from anaemia [1]. This condition is a major public health problem, particularly among children < 5 years of age, in many low- and middle-income countries, including Bangladesh. The latest available national data indicated that more than half of Bangladeshi preschool children suffered from anaemia in 2011 [2].

Anaemia has severe health and economic consequences. In the short term, it causes fatigue, tachycardia, and breathlessness [3], while in the long term, it negatively affects cognitive and motor development and decreases school performance, productivity in adult life, quality of life, and the employment prospects of affected individuals [4]. Anaemia is also associated with childhood immunity. It decreases children’s’ ability to fight infections, which increase childhood morbidity and mortality [5,6]. A recent paper reported that anaemia is responsible for 5–18% of mortality among children under 5 years of age in Africa [6].

Children in low-income settings become anaemic for several reasons, including iron-deficient diets [7] and recurrent or chronic intestinal parasite infections [8]. Intestinal parasite infections due to soil-transmitted helminths (STHs) contribute significantly to iron deficiency by inducing chronic intestinal blood loss. For example, hookworms discharge anti-clotting factors (i.e., coagulase, a blood thinner) that promote a constant blood loss [9,10].

The prevalence of STH infections in low-income settings is very high, despite global initiatives to eliminate STH infection-related morbidity in children by 2030 [11]. Worldwide, more than 1.5 billion people (24% of the human population) are infected with STHs, and more than 80% of these infections occur in low-income countries [11,12]. A previous study has estimated the prevalence of STH infection at 34.6% in Bangladesh [13]. Recommended control measures for STH infections have included deworming or mass drug administration (MDA) with albendazole or mebendazole to children in endemic countries [14]. A recent post-intervention survey conducted in 10 districts of Bangladesh revealed a significant decline to 14% for the prevalence of STH infections in 2017, versus a prevalence of 79.8% in 2005 [15]. This substantial reduction in STH infections among children was attributed to deworming. Therefore, this intervention would also be expected to decrease the prevalence of anaemia.

Several studies based on randomised controlled trials, systematic reviews, and meta-analyses have found mixed results regarding the association between deworming and the prevalence of anaemia among children. A randomised controlled trial conducted in Tanzania reported significant improvement in the mean haemoglobin concentrations among the treatment group from the baseline time point to the time of the 24-month follow-up [16]. A systematic review conducted in 2018 reported a significant decrease in anaemia prevalence and the burden of iron deficiency anaemia among school-aged children due to deworming [17]. However, the effects of deworming on anaemia depend on several other factors, including the duration of the deworming programme, follow-up after deworming, and integration of the deworming programme with other health, hygiene, and nutrition interventions [17]. Another systematic review conducted for children < 5 years of age in Ethiopia observed a significantly higher risk of anaemia if the children were not dewormed [18]. However, these results are not consistent with another study based on the Cochrane Database of Systematic Reviews that concluded that single or multiple deworming drugs may have few or no effects on the average haemoglobin level of the study participants [19], which may be due to rapid worm infection rebound after deworming.

These findings about the effectiveness of deworming on anaemia prevalence have led to the suggestion of using combined interventions that include the use of periodic deworming medicine and supplementation with multiple micronutrient powders (MNPs), as these two interventions are likely to complement each other [20,21]. MNPs are small sachets containing five micronutrients (iron, zinc, folic acid, and vitamins A and C). The recommended use is one sachet per day (mixed with regular meals) to improve the micronutrient levels in the regular diets of children. The reductions in anaemia prevalence among children administered MNPs regularly has been widely established through several systematic reviews and meta-analyses [21,22]. However, most of the articles assessed in those reviews were implemented as small-scale controlled studies; therefore, limited evidence is available to support the effectiveness of MNP supplements in preventing childhood anaemia when implemented at a national scale. Implementation of MNP supplementation at this broader scale may demonstrate better effectiveness when programmes consider other nutrition-sensitive interventions, including deworming. A study conducted in Bangladesh that used home fortification with MNPs together with anthelmintic interventions demonstrated a significant reduction in anaemia prevalence among children < 5 years of age [20]. However, this study was also implemented in a small-scale, controlled setting, and the authors stressed that the study’s design feature limited the policy implications of the study and that an urgent need existed for evidence of the effectiveness of this combined intervention at the national scale.

Bangladesh is one of the pioneer countries that has started both mass deworming and nationwide MNP home fortification interventions. This programme began decades ago, with deworming intervention initially implemented through the observation of a National Deworming Day and later celebrated as a biannual Deworming Week. The programme provided a single dose of mebendazole (a 500 mg tablet) to children aged 5–12 years across the whole country, and the reported coverage was very high despite some concerns regarding reporting bias [23]. In addition to these national programmes, Bangladeshi physicians, paramedics, and local non-formal doctors also prescribe deworming drops and syrups for infants and young children (aged six months or above since they have started complementary feeding) for diagnosis with or susceptibility to STH infections.

One previous article has also documented a high coverage of MNP home fortification [24]. However, the same study also reported that the regular use of MNPs (i.e., the effective coverage) was very low, indicating that substantial effort is needed in the programme to increase the effective MNP coverage [24]. Conversely, the effectiveness of both interventions in combination has not been well documented, but proof of effectiveness may incentivise national policies towards further investments in escalating coverage of both these interventions. Thus, measurements of the combined effectiveness of deworming and MNP interventions on childhood anaemia are essential among children aged 6–59 months. The aim of the present study was to assess the effects of combining deworming and MNP home fortification on the prevalence of anaemia among children < 5 years of age in Bangladesh after adjusting for sociodemographic factors.

## 2. Materials and Methods

### 2.1. Study Design and Settings

We used data from five cross-sectional surveys conducted to evaluate the Maternal Infant and Young Child Nutrition (MIYCN) program implemented by BRAC (formerly known as Bangladesh Rural Advancement Committee) in three BRAC programme platforms: the Maternal, Neonatal and Child Health (MNCH) programme platform, the Alive and Thrives (A&T) platform, and the BRAC Nutrition platform. The aim of the MIYCN programme was to reduce anaemia prevalence among under-five children. Interventions related to home fortification with MNP were implemented by the BRAC community health workers (CHWs) across all programme areas. However, no intervention on deworming was provided as part of the MIYCN programme. The participating children received deworming medication only when they visited physicians or local doctors when unwell. Detailed descriptions of the survey designs and BRAC programme platforms are published elsewhere [24,25,26]. Data for the present study were collected from 27 districts across eight administrative divisions (Barisal, Dhaka, Chittagong, Khulna, Mymensingh, Rajshahi, Rangpur, and Sylhet divisions) and from two urban slums in the Dhaka North City-Corporation in Bangladesh.

### 2.2. Study Population and Sampling

The study population included children aged 6–59 months and their caregivers. The caregivers were mostly biological mothers, grandmothers, and aunts of the children. We identified one child from each household. If more than one eligible child lived in a selected household, we randomly selected one as the study subject. The households were selected using a two-stage sampling strategy. In the first stage, we identified the primary sampling unit (the study community, which was the catchment area of the BRAC CHW) through systematic random sampling. In the second stage, we used a mapped segmented sampling strategy to identify the study households. A detailed description of the sampling strategy is published elsewhere [26].

### 2.3. Data Collection

We used a structured questionnaire and conducted face-to-face interviews with caregivers to collect the primary data. The survey questionnaire was validated through real-world field testing before it was finalized and transferred to an electronic survey platform for data collection. Eight data collection teams were engaged in each iteration of the surveys. Each team consisted of two interviewers, a medical technologist, and a supervisor of the team. Every member of the data collection team had been trained in survey interviews, the content of the questionnaire, collection of blood samples, measuring haemoglobin, and topics on research ethics. We collected capillary finger-prick blood samples to measure haemoglobin. The trained medical technologist collected blood samples from the children and used a second drop of blood for the assessment. At first, the medical technologist wiped away the first drops of blood using a dry gauze pad, gently pressed the finger until the second drop of blood appeared and ensured that the second drop was sufficient to fill the microcuvette completely. We measured haemoglobin levels using a HemoCue machine (Hb 301, HemoCue AB, Angelholm, Sweden).

### 2.4. Outcome Variables

The outcome variable of this study was the prevalence of anaemia among children aged 6–59 months. We defined anaemia as a haemoglobin concentration of <11.0 g/dL [27].

### 2.5. Exposure and Other Covariates

The exposure variables were the coverage of deworming and the regular use of MNP. We asked caregivers whether the participant children had taken any deworming medicine in the six months prior to the survey. The responses considered were ‘yes’, ‘no’, ‘don’t know’ or ‘can’t remember’. The coverage of regular use of MNP was documented by a proxy indicator, called the MNP effective coverage, which indicated the use of MNP for at least three of the seven days immediately preceding the survey. We asked the caregiver how many MNP sachets had been used for the participant child in the past seven days. Based on the caregiver’s response, we categorised the effective MNP coverage as ‘yes’ (if the child used ≥3 sachets) or ‘no’ (if the child used <3 sachets).

Other covariates included the child’s age (in months) and sex (male or female), the caregiver’s age (in years) and education (completed years of schooling, categorised as <5 years or ≥5 years of schooling), the father’s age (in years) and education (completed years of schooling, categorised as <5 years or ≥5 years of schooling), the household size (number of household members, categorised as <5 members or ≥5 members), the household toilet facilities (categorised as households with improved or unimproved toilet facilities), the study sites (the BRAC programme platforms), and the household wealth index based on household assets (categorised as poor, middle, and rich). We performed a principal component analysis to calculate the household wealth index based on household items such as the materials used for the floor, roof, and wall of the house and household assets. The five surveys were categorised as survey-1, survey-2, survey-3, survey-4, and survey-5 according to the time they were conducted.

### 2.6. Ethical Considerations

The evaluation protocol, which included all five surveys, was approved by the Institutional Review Board of icddr,b, Bangladesh. Prior to collecting the data, the interviewers explained the study objectives and procedures to the caregivers and took their written informed consent for participating in the study. All participation was voluntary. Before analysing the data, we removed all identifying information on participants from the data set.

## 3. Analysis

We first used descriptive statistics to describe the basic characteristics of the key variables. We calculated means and standard deviations for all continuous variables, including the ages of the children, caregivers, and fathers and the father’s monthly income. We also estimated frequencies and proportions for all categorical variables and compared them between the groups. We assessed the relationships between anaemia prevalence and the exposure variables by performing cross tabulations, and we distributed the anaemia prevalence across three exposure categories: the MNP effective coverage, deworming, and the combined MNP effective coverage and deworming (MNP + Deworming). We then performed a Pearson chi-squared test to identify any significant differences between the exposed and unexposed groups (who had not been exposed to either deworming or MNP + Deworming). We performed binary logistic regression and calculated unadjusted odds ratios (ORs) to understand whether any association existed between the outcome variable (prevalence of anaemia) and the sociodemographic characteristics of the study participants. We performed a similar analysis to assess the association between anaemia prevalence and the exposure variables. We found significant associations between the outcome and exposure variables; therefore, we considered two exposure variables—deworming and MNP + Deworming—and we then calculated adjusted odds ratios (AORs) to measure whether the exposures had any significant associations with anaemia prevalence. In this model, we adjusted for all the variables that were identified as being significantly associated with outcomes in the unadjusted model. We adjusted sample design effects by adjusting for the clusters (primary sampling units) while performing the regression analysis. We also checked the multicollinearity among the independent variables and found an average variance inflation factor 1.27, which indicated negligible collinearity existed among the independent variables. Finally, we performed a Pearson’s goodness-of-fit (GoF) test to determine the adequacy of the model. The GoF estimated a Pearson χ^2^ value of *p* = 0.110, indicating a reasonably good fit for the model. A *p*-value of <0.05 was considered statistically significant for all tests. Analysis was performed using STATA version 16.1 (Stata Corporation, College Station, TX, USA).

## 4. Results

A total of 9948 caregiver cases were considered for this paper. Table 1 describes the background characteristics of the study participants. The mean age of the children was 30 months, and slightly more than half of them (52%) were male. The mean ages of the caregivers and fathers were 27 and 34 years, respectively. More than 79% of the caregivers and 67% of the fathers had completed at least five years of education. About 59% of the households had five or more household members, and about one third of the households were poor based on the household wealth index. Most of the households (94%) had improved toilet facilities. More than half of the study participants were from the A&T programme areas (51%), followed by the Nutrition (34%) and MNCH (15%) platforms. In terms of survey time points, 28% of the respondents participated in survey-1 and 23% in survey-4. Approximately 8% of the children had effective MNP coverage, and 45% of the children had received deworming medication within six months of the surveys; however, the percentage of children receiving combined coverage of both effective MNP and deworming (MNP + Deworming) was only 3.2% (n = 318).

Table 2 describes the anaemia prevalence and percentage of children who received deworming medication and MNP effective coverage by age group. Younger children were less likely to receive deworming and MNP effective coverage compared with their older counterparts (Table 2). Anaemia prevalence was highest among those aged 12 to 23 months and was similar among other age groups. Figure 1A–C) shows the distribution of anaemia prevalence by deworming status and effective MNP coverage. No significant difference was detected between the effective MNP coverage and anaemia prevalence (Figure 1A), but the differences between deworming and anaemia prevalence and between the MNP + Deworming condition and anaemia prevalence (Figure 1B,C) were highly significant (*p* < 0.001). Similar results were observed for the unadjusted logistic regression model (Table 2). The children who received deworming were 43% (OR 0.57; 95% CI 0.53, 0.63; *p* < 0.001) less likely to suffer from anaemia compared with the children who did not receive deworming. The children who had both MNP and deworming coverage had 53% (OR 0.47; 95% CI 0.35, 0.61; *p* < 0.001) less chance of being anaemic compared with the children who did not have both MNP and deworming coverage (Table 3).

The unadjusted model (Table 3) revealed significant associations between several independent variables and anaemia prevalence. The age of the children, caregivers, and fathers were all significantly associated (*p* < 0.001) with the prevalence of anaemia (i.e., as the child’s age increased, the prevalence of anaemia decreased significantly). For example, the anaemia prevalence decreased by 4% if the child’s age increased by one month (OR 0.96; 95%CI 0.957, 0.964; *p* < 0.001). Fathers and caregivers with five and more years of schooling were less likely to have anaemic children than illiterate parents and parents with less than five years of schooling. The children in households in the middle or wealthier categories in the household wealth index were significantly less likely to be anaemic than the children in households in the poor household category. Similar associations were observed when children participated from the A&T and Nutrition programme platforms. In terms of survey timepoints, children who participated in survey-2 and survey-3 were significantly more likely to be anaemic than were children who participated in survey-1 (Table 3).

Table 4 shows the adjusted effects of deworming and the combined MNP + Deworming on the prevalence of anaemia. Deworming alone was significantly associated with anaemia prevalence after adjusting for the independent variables that were significantly associated with anaemia prevalence in the unadjusted model. For example, after adjusting for child age, caregiver age, caregiver education, father age, father education, household wealth index, and study timepoints, the children who received deworming medicine within the six months prior to the surveys were 13% less likely to be anaemic compared with the children who did not receive deworming (AOR 0.87; 95% CI 0.79, 0.97; *p* = 0.009). In the adjusted model, the combined effect of deworming and effective MNP coverage was also significantly associated with the prevalence of anaemia. The children who were exposed to both deworming and effective MNP coverage were 30% less likely to be anaemic compared with the children who were unexposed to combined MNP + Deworming (AOR 0.70; 95% CI 0.52, 0.94; *p* = 0.018 (Table 4).

## 5. Discussion

Overall, the study findings revealed that combined interventions involving regular use of MNPs and periodic deworming medications would have positive effects in reducing anaemia prevalence among children < 5 years of age in Bangladesh. These results are very promising for low-income countries, and particularly for those countries that have a high burden of anaemia and are struggling to find appropriate interventions to address the high prevalence of anaemia among young children. Both home fortification with MNPs and deworming are very low-cost interventions and may be affordable in many countries. The cost for a box of 30 MNP sachets was cited as USD 0.39 [28], and the price of one dose of deworming medication can be as low as USD 0.02 [17]. By contrast, anaemia imposes significant clinical and economic burdens across many parts of the world. Anaemia can also substantially accelerate the progression of several chronic and infectious diseases, further increasing the treatment cost [29]. A previous study calculated average annual costs per patient with anaemia at USD 14,535, or 54% higher than the costs for non-anaemic patients [30].

Our findings suggested that deworming against STH infections can be viewed as a protective effort against anaemia. Evidence in the global literature also found that STH infections are consistently identified as strong risk factors for iron-deficient anaemia in children [31,32]. Bangladesh has all the risk factors for worm infections, including its tropical climate, which provides a favourable environment to accelerate all parasitic diseases and to present a large burden of these infections. Worms are generally transmitted through eggs present in human faeces, and the soil in areas where people have poor sanitation facilities is often contaminated with eggs. One study determined a prevalence of STH eggs in household soils in Bangladesh of 78%, which was significantly higher (*p* < 0.001) compared with the prevalence in household soils in Kenya [33]. The prevalence of soil ingestion by young children is very high; for example, a recent study estimated that children ingested an estimated 162–234 mg dry soil/day [34], which may be sufficient for young children to become infected with several STHs. In Bangladesh, all 64 districts are considered to be high STH endemic zones, with an estimated burden of 78 million infections [22]. Thus, the country must continue with its mass deworming interventions for children in rural areas, as those children are at particularly high risk of STH infections.

Our study findings indicate a substantially lower anaemia presence among the children who used the MNP supplement with homemade foods and had used deworming medication in the six months before the survey. Deworming may have independent effects on anaemia, but adding deworming with multiple micronutrients that include iron would significantly increase the intervention effectiveness and substantially reduce the prevalence of anaemia. The reduction rates in the anaemia prevalence were 13% when children had only the deworming treatment but were 30% when the children received both deworming and the MNP supplement. This emphasised the importance of integrating interventions as combinations of both nutrition-specific and nutrition-sensitive interventions to be taken in place. Treating STH infections can be an important first step to child health, but a rapid recovery from iron deficiency could require an integrated intervention with deworming and multiple micronutrient supplements [35]. Due to worm infections, substantial amounts of micronutrients are lost, with iron being the key depleted nutrient [35]. Immediately after deworming, young children have difficulty in recovering the nutrients lost from the diet, as the worm infections make them physiologically weak and incapable of absorbing the recommended nutrients from their diet. Red meat is considered a good source of iron, but it is a costly food for people in low-income settings. On the other hand, a recent study conducted in Indonesia reported no effect of meat consumption on the prevalence of anaemia [36]. The bioavailability of iron contained in an MNP sachet is more easily absorbed than dietary iron. Thus, a feasible and appropriate intervention is to integrate deworming interventions with multiple micronutrient supplements in countries like Bangladesh.

Our analysis indicated no significant effects of effective MNP coverage on the prevalence of anaemia, which might be surprising in the context of the global literature. A previous systematic review and meta-analysis suggested that home fortification with MNPs significantly reduced anaemia by 18% and iron deficiency by 53% among infants and young children compared with no intervention or a placebo [37]. However, all the studies included in that review were experimental randomised controlled trials and had been implemented in small-scale controlled settings. Only limited evidence suggested that MNP interventions implemented at the national scale demonstrated a better effectiveness against anaemia or iron deficiency among children. Our study evaluated the data of a programme implemented at a scale-up setting across 27 of 64 districts in Bangladesh. A previous study that evaluated nationally scaled-up programmes in Kyrgyzstan reported no significant effects of point-of-use fortification with MNP on the prevalence of anaemia after four years of programme implementation [38]. Similar results were also observed in Uganda, where the interventions were not associated with the prevalence of anaemia, despite the high implementation fidelity demonstrated in the programme [39].

Another key concern identified in our study is the low regular use of MNPs, as this indicates that programme-level constraints are hindering the implementation of MNP interventions at the community level. The BRAC programme is implemented via its volunteer CHWs, and the success of this programme has mostly depended on regular home visits, as the BRAC CHWs sell MNP sachets to the visited households and provide nutrition education to the caregivers. A previous study on BRAC CHWs observed that the children whose caregivers received at least one visit from their CHW in the past 12 months were more than 8 times more likely to have effective MNP coverage than were the children whose caregivers had not received any CHW visits [24]. However, the prevalence of CHW home visits was very low, as less than half the caregivers had received at least one CHW visit in the past 12 months before the survey, and a very low percentage (23%) had received a visit within 2 months of the survey [40]. BRAC should consider incentivising and motivating its CHWs to increase their home visits to ensure high effective MNP coverage.

## 6. Limitations and Strengths of the Study

This study has several limitations. One was that we used data from an evaluation that did not consider a control group, thereby limiting our ability to measure the comparative effectiveness of the deworming and MNP coverage on anaemia. Other limitations were that we depended on recall by the caregivers for collecting information about exposure of the children to deworming medication for the past six months before the survey and the self-reported MNP effective coverage. These might have imposed a recall bias when the caregivers endeavoured to provide accurate information. We resolved this problem by assessing doctors’ prescriptions and product packaging for the respective children who were exposed to deworming medication. However, we did not record data about the proportion of dewormed cases that were actually verified by assessing the prescription or product packaging. Moreover, our paper might have a problem with endogeneity. Despite carefully considering all the covariates and exposure variables, the analysis might have missed additional variables that could have influenced the outcome estimations. A further limitation was the low prevalence of the effective MNP coverage and of the combined exposure to MNP + Deworming; this likely reduced the statistical power when estimating the association between outcome and exposures.

The study also has some strengths. One was our use of pooled data from five surveys, as this allowed us to assess the outcomes with enhanced statistical power and to compare outcomes across different exposures. Another was that the data for the study were collected from 27 districts across Bangladesh; these were well representative of the larger national population, thereby providing confidence regarding the outcome estimations when scaled up in larger population settings.

## 7. Conclusions

The combined effects of deworming and MNP supplementation on the reduction in anaemia prevalence highlighted the importance of using integrated and multidisciplinary intervention strategies. The integration of nutrition-specific and health-related interventions is likely to address the high burden of anaemia among young children, particularly those in low-income settings. The BRAC MIYCN programme should incorporate periodic deworming medication with its MNP interventions for children from their weaning age. Increasing the effective coverage of MNP should also be a key priority of the MIYCN programme. The MNP coverage may be enhanced by addressing current programme-level constraints, which have already been identified in recent studies [40,41]. However, prior to implementation of modified interventions in real-world settings, conducting a feasibility evaluation is essential for assessing the likelihood that the MIYCN programme and BRAC resources can be adapted with the proposed revisions of the implementation strategies and for measuring the outcomes of the revised interventions in real-world settings.

## Figures and Tables

**Figure 1 nutrients-14-00150-f001:**
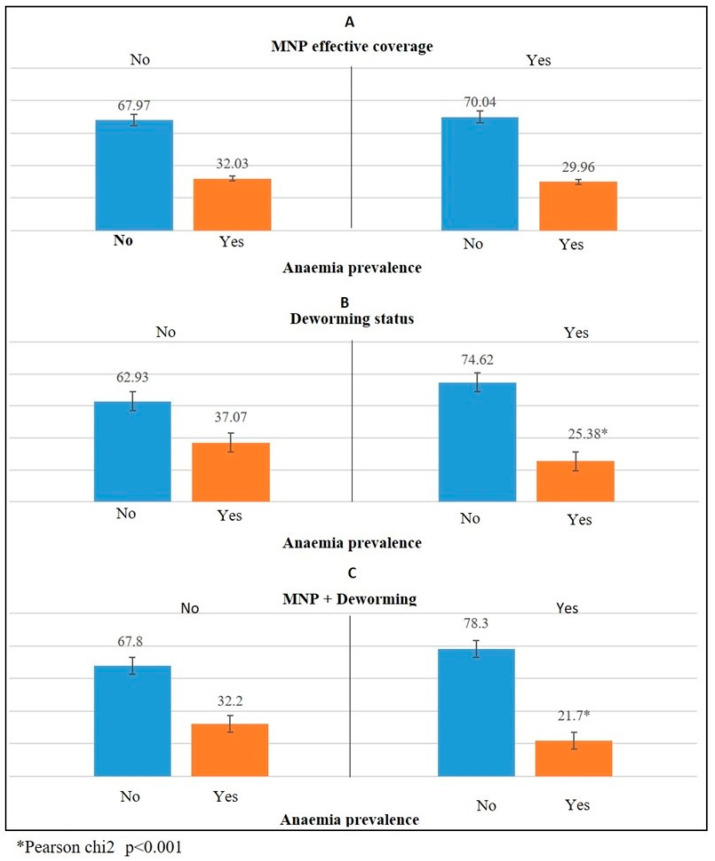
Distribution of anaemia prevalence by deworming status and MNP coverage among children aged 6–59 months. (**A**) MNP effective coverage; (**B**) Deworming status; (**C**) MNP + Deworming.

**Table 1 nutrients-14-00150-t001:** Background characteristics of study participants.

Variables	Estimation
Age of children in months μ (SD)	29.73 (14.52)
Sex of children n (%)	
Male	5199 (52.26)
Female	4749 (47.74)
Caregiver age in years μ (SD)	26.73 (6.01)
Caregiver education n (%)	
<5 years of schooling	2108 (21.19)
≥5 years of schooling	7840 (78.81)
Father age in years μ (SD)	33.53 (6.62)
Father education n (%)	
<5 years of schooling	3264 (32.81)
≥5 years of schooling	6684 (67.19)
Household size (number of household members) n (%)	
<5 members	4091 (41.12)
≥5 members	5857 (58.88)
Household wealth index n (%)	
Poor	3409 (34.27)
Middle	3262 (32.79)
Rich	3277 (32.94)
Household with n (%)	
Unimproved toilet facilities	550 (5.54)
Improved toilet facilities	9384 (94.46)
Study sites (BRAC program platform) n (%)	
MIYCN area	1527 (15.35)
A&T Program area	5058 (50.84)
Nutrition program area	3363 (33.81)
Study time-points n (%)	
Survey-1	2758 (27.72)
Survey-2	1979 (19.89)
Survey-3	1527 (15.35)
Survey-4	2300 (23.12)
Survey-5	1384 (13.91)
Effective coverage of MNP n (%)	
No	9157 (92.05)
Yes	791 (7.95)
Deworming in the 6 months prior to the survey n (%)	
No	5516 (55.45)
Yes	4432 (44.55)
Combined: MNP effective coverage and deworming n (%)	
No	9630 (96.80)
Yes	318 (3.20)

SD—standard deviation; MNP—micronutrient powders; MIYCN—Maternal Infant and Young Child Nutrition, A&T—Alive and Thrives.

**Table 2 nutrients-14-00150-t002:** Percentage of children who were anaemic, received deworming medication, and MNP effective coverage by age group.

Age in Months	Prevalence of Anaemia n (%)	Coverage of Deworming n (%)	Effective Coverage of MNP n (%)
6 to 11	689 (21.74)	39 (0.88)	130 (16.43)
12 to 23	1080 (34.07)	813 (18.34)	293 (37.04)
24 to 35	669 (21.10)	1406 (31.72)	183 (23.14)
36 to 59	732 (23.09)	2174 (49.05)	185 (23.39)

**Table 3 nutrients-14-00150-t003:** Unadjusted odds ratio to assess effects of socio-demographic variables, combined coverage of deworming, and MNP coverage on the prevalence of anaemia.

Independent Variables	OR (95% CI)	*p*-Value
Age of children in months	0.96 (0.957, 0.964)	<0.001
Sex of children		
Male	-	
Female	0.996 (0.917, 1.082)	0.919
Caregiver age in years	0.98 (0.97, 0.98)	<0.001
Caregiver education		
<5 years of schooling	-	
≥5 years of schooling	0.89 (0.81, 0.99)	0.032
Father age in years	0.98 (0.98, 0.99)	<0.001
Father education		
<5 years of schooling	-	
≥5 years of schooling	0.91 (0.83, 0.99)	0.047
Household size (number of household members)		
<5 members	-	
≥5 members	1.09 (0.99, 1.18)	0.068
Household wealth index		
Poor	-	
Middle	0.80 (0.72, 0.89)	<0.001
Rich	0.77 (0.69, 0.87)	<0.001
Household with		
Unimproved toilet facilities		
Improved toilet facilities	0.97 (0.79, 1.18)	0.736
Study sites (BRAC program platform)		
MIYCN area	-	
A&T Program area	0.67 (0.58, 0.77)	<0.001
Nutrition program area	0.80 (0.69, 0.94)	0.005
Survey time-points		
Survey-1	-	
Survey-2	1.63 (1.38, 1.92)	<0.001
Survey-3	1.57 (1.33, 1.85)	<0.001
Survey-4	1.12 (0.97, 1.31)	0.131
Survey-5	0.84 (0.71, 1.02)	0.052
Effective coverage of MNP		
No		
Yes	0.91 (0.77, 1.08)	0.263
Deworming in the 6 months prior to the survey		
No	-	
Yes	0.58 (0.53, 0.63)	<0.001
Combined coverage: MNP effective coverage and deworming (MNP + Deworming)		
No	-	
Yes	0.58 (0.44, 0.77)	<0.001

SD—standard deviation; MNP—micronutrient powders; MIYCN—Maternal Infant and Young Child Nutrition, A&T—Alive and Thrives.

**Table 4 nutrients-14-00150-t004:** Effects of combined coverage of MNP effective coverage and deworming on the prevalence of anaemia.

Exposures	AOR (95% CI)	*p*-Value
Deworming in the 6 months prior to the survey		
No	-	
Yes	0.87 (0.79, 0.97)	0.009
Combined: MNP effective coverage and deworming (MNP + Deworming)		
No	-	
Yes	0.70 (0.52, 0.94)	0.018

Model adjusted for child age, caregiver age, caregiver education, father age, father education, household wealth index, and study timepoints.

## Data Availability

The data presented in this study are available on request from the corresponding author.

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
