# Peer review of "The Effects of Deworming and Multiple Micronutrients on Anaemia in Preschool Children in Bangladesh: Analysis of Five Cross-Sectional Surveys"

_nutrients, 2021, doi:10.3390/nu14010150_

Round 1
Reviewer 1 Report
The authors used a cross-sectional study design to assess the effects of combining deworming and MNP home fortification on the prevalence of anaemia among children <5 years of age in Bangladesh. The topic is interesting. However, I have several questions:
- Was the questionnaire of exposure assessment (i.e., coverage of deworming and the regular use of MNP) validated?
- Did the author collected maternal age at birth and maternal education, if so, please also adjust them.
- did the author consider the effect of altitudes when defining anaemia?
- The potention effect of premature birth should also be considered
- What's the proportion of iron fortification in the study population?
Author Response
Reviewer 1
The authors used a cross-sectional study design to assess the effects of combining deworming and MNP home fortification on the prevalence of anaemia among children <5 years of age in Bangladesh. The topic is interesting. However, I have several questions:
1. Was the questionnaire of exposure assessment (i.e., coverage of deworming and the regular use of MNP) validated?
Our responses: The survey questionnaire including items to assess both the exposures: coverage of deworming and the regular use of MNP has been validated through real-world field testing before it was finalized and transferred to an electronic survey platform. We have added this statement in the data collection section, lines 144-146.
2. Did the author collected maternal age at birth and maternal education, if so, please also adjust them.
Our responses: Unfortunately, we did not collect data on maternal age at birth, but collected caregiver education – which may be a proxy to maternal education as more than 95% of caregivers are biological mothers of the children who participated in this study. We adjusted caregivers’ education in our analysis.
3. Did the author consider the effect of altitudes when defining anaemia?
Our responses: We used WHO cut-off points for defining anemia, which has been adjusted for altitudes for the children in low-income countries, including for children in Bangladesh.
4. The potential effect of premature birth should also be considered
Our responses: This may be an important indicator to be considered in the analysis, however, we did not collect data related to premature birth.
5. What's the proportion of iron fortification in the study population?
Our responses: Approximately 8% of the children had effective MNP coverage (reported in line 227), which may be considered as a proportion of children who received iron-based micronutrient fortification. We do not have information on any other iron fortified foods the children may have consumed.
Reviewer 2 Report
Paper title: “The effects of deworming and multiple micronutrients on anaemia in preschool children in Bangladesh: analysis of five cross—sectional surveys”
Thank you for your paper, I do believe it will make a valuable contribution to understand the effects of deworming and multiple micronutrients on anaemia in preschool children in Bangladesh. Your paper is well written, and interesting. However I do suggest some changes and points to address below:
Background:
I think the second paragraph about the consequences of anemia can expand some literature review. The analysis of the causes of anemia in the second paragraph can be moved to the next paragraph. One paragraph only needs to talk about one thing.
Endogeneity problem:
Endogeneity is a major concern in assessing the effects of deworming and micronutrients on anaemia rates in children. On the one hand, there may be some unobserved factors associated with the children anaemia outcome that may also influence the child anaemia rate; On the other hand, there could be reverse causality, with parents more likely to choose deworming or micronutrient supplements if they think their child is anaemic. Although deworming and trace elements can affect the anemia rate after controlling for observable factors, they cannot solve the endogeneity problem, and I think the author can try to use more methods to solve this problem.
Materials and Methods:
In part 2.6, I think you should explain clearly the assignment of the variable-'Deworming in the past 6 months of survey’, specifically, when the caregivers replied ‘don't know’ or ‘can't remember’, how did the author handle this and assign a value to this variable?
In the meanwhile, for the variable-Household wealth index, maybe you should tell the reader how you classify it.
Statistical analysis:
I think the author could have used more robustness tests to make the results more convincing, such as heterogeneity analysis.
Author Response
Paper title: “The effects of deworming and multiple micronutrients on anaemia in preschool children in Bangladesh: analysis of five cross—sectional surveys”
Thank you for your paper, I do believe it will make a valuable contribution to understand the effects of deworming and multiple micronutrients on anaemia in preschool children in Bangladesh. Your paper is well written, and interesting. However, I do suggest some changes and points to address below:
Background:
- I think the second paragraph about the consequences of anemia can expand some literature review. The analysis of the causes of anemia in the second paragraph can be moved to the next paragraph. One paragraph only needs to talk about one thing.
Our responses: As suggested we have expanded the paragraph with additional literature and separated it from causes of anemia. See lines 40-43.
Endogeneity problem:
- Endogeneity is a major concern in assessing the effects of deworming and micronutrients on anaemia rates in children. On the one hand, there may be some unobserved factors associated with the children anaemia outcome that may also influence the child anaemia rate; On the other hand, there could be reverse causality, with parents more likely to choose deworming or micronutrient supplements if they think their child is anaemic. Although deworming and trace elements can affect the anemia rate after controlling for observable factors, they cannot solve the endogeneity problem, and I think the author can try to use more methods to solve this problem.
Our response: We agreed and thanks for this important comment. Despite we have carefully considered all the covariates and exposure variables, there might have other variables that could influence the outcome of exposures. We mentioned it in the discussion, under limitations (page 10, lines 386-388).
Materials and Methods:
- In part 2.6, I think you should explain clearly the assignment of the variable-'Deworming in the past 6 months of survey’, specifically, when the caregivers replied ‘don't know’ or ‘can't remember’, how did the author handle this and assign a value to this variable?
Our responses: We further reviewed the questionnaire and the response option was only ‘yes’ or ‘no’. Now we corrected the sentence in the revised manuscripts, lines 163-164.
- In the meanwhile, for the variable-Household wealth index, maybe you should tell the reader how you classify it.
Our responses: We performed a principal component analysis to calculate the household wealth index based on household items such as the materials used for the floor, roof, and wall of the house, and household assets. We added this sentence in method section, lines 184-186.
Statistical analysis:
- I think the author could have used more robustness tests to make the results more convincing, such as heterogeneity analysis.
Our responses: To our knowledge, heterogeneity analysis or heterogeneity test is performed in meta-analysis to see the proportions of dissimilarities among different studies. In this study, we included five-cross sectional surveys from a single evaluation project, all the surveys are identical to each other as they used the same sampling framework and questionnaire. Thus, we think the heterogeneity test is not applicable for this study. In this paper, we checked whether the analysis has any problem with multicollinearity, among the independent variables and found an average variance inflation factor 1.27, which indicated negligible collinearity existed among the independent variables. We also performed Pearson’s Goodness-of-fit (GoF) to test for model diagnosis (see page 5, lines 214-218).
Reviewer 3 Report
This paper bundles data from five cross-sectional studies conducted in Bangladesh to examine the effects of deworming and taking multiple micronutrient powders on anemia in children under five. It assesses the relevance of the intervention for children under five.
Line 100-103, page 3
“Bangladeshi physicians, paramedics and local non-formal doctors also prescribe deworming drops and 101 syrups for infants and young children (aged six months or above since they started complementary feeding) for diagnosis with or susceptibility to STH infections.”
Does this mean that in this study, no anthelmintics were prescribed to infants less than 6 months after the start of complementary diet?
Line 133, page 3
“The study population included children aged 6–59 months and their caregivers.”
I am wondering whether the infants who had been on complementary diet for less than 6 months in this study were not prescribed anthelmintics at all. Or, if they were, the number is probably very small. It would be better to exclude this age group from the study of the effect of anthelmintics on anemia. If you redo the analysis, could your conclusions change?
Line 247-249, page 7
"For example, the anaemia prevalence decreased by 4% if the child’s age increased by one month (OR 0.96; 95%CI 0.957, 0.964; p<0.001)."
Are there any previous reports or estimation on age-specific incidence of soil-transmitted helminths or parasites among children under 5 years old in Bangladesh? Since this paper states that the prevalence of anemia is age-dependent, I thought it would be good to have a comparison with the age-specific incidence of soil-transmitted helminths or parasites. It would also be good to show the percentage of deworming medication taken by age group.
Author Response
This paper bundles data from five cross-sectional studies conducted in Bangladesh to examine the effects of deworming and taking multiple micronutrient powders on anemia in children under five. It assesses the relevance of the intervention for children under five.
- Line 100-103, page 3: “Bangladeshi physicians, paramedics and local non-formal doctors also prescribe deworming drops and syrups for infants and young children (aged six months or above since they started complementary feeding) for diagnosis with or susceptibility to STH infections.” Does this mean that in this study, no anthelmintic were prescribed to infants less than 6 months after the start of the complementary diet?
Our responses: Our study only included children aged 6 months to 59 months at the time of data collection. In our analysis, 39 children less than 12 months (or less than 6 months after the start of the complementary feeding) have received deworming and 130 children were classified as receiving MNP effective coverage. We have now added Table 2 that describes the proportion of younger children who received deworming by age groups.
- Line 133, page 3: “The study population included children aged 6–59 months and their caregivers.” I am wondering whether the infants who had been on complementary diet for less than 6 months in this study were not prescribed anthelmintics at all. Or, if they were, the number is probably very small. It would be better to exclude this age group from the study of the effect of anthelmintics on anemia. If you redo the analysis, could your conclusions change?
Our responses: Thanks for this suggestion. As responded above, we did a subset analysis excluding the children aged less than 12 months. However, the combined effect of deworming and MNP on anemia prevalence did not change much, differences are only 3% points (AOR 0.70, 95% CI: 0.53-0.92 vs. 0.73 95% CI: 0.55-0.98) rather it substantially reduced statistical power in the analysis. Thus, we have decided to keep the original analysis considering potential policy implications to engage younger children in future interventions as the children in this group are vulnerable to both high prevalence of anemia and STH infections.
- Line 247-249, page 7: "For example, the anaemia prevalence decreased by 4% if the child’s age increased by one month (OR 0.96; 95%CI 0.957, 0.964; p<0.001)." Are there any previous reports or estimations on the age-specific incidence of soil-transmitted helminths or parasites among children under 5 years old in Bangladesh? Since this paper states that the prevalence of anemia is age-dependent, I thought it would be good to have a comparison with the age-specific incidence of soil-transmitted helminths or parasites. It would also be good to show the percentage of deworming medication taken by age group.
Our responses: This is also an important point for the perspectives of this paper. Unfortunately, we did not find any reports through our literature review, that described the age-specific incidence of soil-transmitted helminths or parasites among children under 5 years old in Bangladesh.
We have added a table (Table 2) that describes the percentage of deworming medication taken by age group.
Round 2
Reviewer 1 Report
The authors have issued all the questionas that I have raised. I have no further comments.
Reviewer 2 Report
The author have edited the paper. So I don't have any other comments. Thanks.
Reviewer 3 Report
Line 258, page 6: Thank you for adding Table 2. Furthermore, adding the prevalence of anaemia in each age group to Table 2 would contribute to the reader's understanding of the study.
Line 266, page 7: "For example, the anaemia prevalence decreased by 4% if the child’s age increased by one month (OR 0.96; 95%CI 0.957, 0.964; p<0.001)." Is the high prevalence of anaemia at younger ages, especially 6-11 months, due to parasites? Is it possible that most of the anaemia in 6-11 months is nutritional iron deficiency anaemia? If it is predominantly nutritional iron deficiency anaemia, wouldn't it create a bias to include this age group, where anthelmintic use is extremely low, in the analysis to determine the efficacy of anthelmintics?
Author Response
Line 258, page 6: Thank you for adding Table 2. Furthermore, adding the prevalence of anaemia in each age group to Table 2 would contribute to the reader's understanding of the study.
Our responses: We have revised Table 2 and the texts in the results sections, line 244-247. We have now added the anemia prevalence as a new column in this Table 2.
Line 266, page 7: "For example, the anaemia prevalence decreased by 4% if the child’s age increased by one month (OR 0.96; 95%CI 0.957, 0.964; p<0.001)." Is the high prevalence of anaemia at younger ages, especially 6-11 months, due to parasites? Is it possible that most of the anaemia in 6-11 months is nutritional iron deficiency anaemia? If it is predominantly nutritional iron deficiency anaemia, wouldn't it create a bias to include this age group, where anthelmintic use is extremely low, in the analysis to determine the efficacy of anthelmintics?
Our responses: We thank the reviewer for further comments on this issue. We don’t know what was the main causes of anemia among the younger children, they are anemic probably due to recurrent parasites infections or iron deficiency, or maybe due to both.
We don’t agree that including this age group would impose bias in our analysis. Younger children are highly susceptible to becoming infected with parasites due to their household contexts as most household floors in rural Bangladesh are made with earth/soil. A previous study (reference # 33) estimated that the prevalence of STH eggs in household soils in Bangladesh was 78%, and another study (reference # 34) estimated soil ingestion by the children from 6 to 11 months was 224 dry mg/day, which was significantly higher than the children in other age groups (12–47 months) average soil ingestion (193 dry mg/day). Thus, children in this younger age group are highly vulnerable to becoming infected with soil-transmitted helminths. Unfortunately, due to inadequate policy and guidelines, children in this age group did not receive anthelmintic treatment unless they were clinically diagnosed. That was possibly the main reason that only a small proportion of children in this age group received anthelmintic treatment. Considering the above evidence and potential policy implications, we would like to keep children 6–11 months in our analysis. We hope the above explanations satisfy reviewer concerns in this regard.